# The Effect of Sintering Temperature on Vickers Microhardness and Flexural Strength of Translucent Multi-Layered Zirconia Dental Materials

Bashayer Alfahed [1,2,*] and Abdullah Alayad [1]

1 Department of Restorative Dental Sciences, College of Dentistry, King Saud University, P.O. Box 60169, Riyadh 11545, Saudi Arabia
2 Department of Clinical Dental Sciences, College of Dentistry, Princess Nourah bint Abdulrahman University, P.O. Box 84428, Riyadh 11671, Saudi Arabia
* Correspondence: bashayer.b.a@live.com

**Abstract:** This study evaluated the effects of the sintering temperature on Vickers microhardness and three-point flexural strength values of two multi-layered zirconia materials. Multi-layered zirconia systems with four distinct layers were selected: DD cube ONE ML (4Y-TZP) and DD cubeX$^2$ ML (5Y-TZP). In total, 96 plate-shaped A2-shade specimens were obtained using individual layers of these two zirconia materials. The individual layers were then divided equally into batches with three different sintering temperatures (1300, 1450, and 1600 °C), and the Vickers microhardness was assessed. Another group of 72 bar-shaped specimens was prepared from the same materials. These were similarly divided into three different sintering temperatures, and the flexural strength was assessed. SEM was used to conduct fractographic analyses. The data were analyzed using SPSS 24.0 software with a *p*-value < 0.05. The microhardness and flexural strength of 4Y-TZP were higher than those of the 5Y-TZP at all the sintering temperatures. A significant difference was found in the microhardness and flexural strength values between groups sintered at different sintering temperatures ($p < 0.05$). The highest microhardness and flexural strength values were found at 1450 °C ($p < 0.05$). The microhardness values of different layers were not significantly different ($p > 0.05$). The sintering temperature and type of ceramic material significantly affected the microhardness and flexural strength. However, the layers did not significantly affect the microhardness.

**Keywords:** zirconia; multi-layered; Vickers microhardness; flexural strength; fractography





## 1. Introduction

Zirconia dental materials are widely used in dental clinics because of their outstanding mechanical and physical properties [1,2]. Manufactures have been striving to meet the high demand for esthetic zirconia restorations. Many techniques have been employed to manufacture highly esthetic zirconia materials. First, the opacity of zirconia materials was improved by increasing the yttria content and thereby increasing the cubic phase, which improved the translucency. However, this was associated with low mechanical performance [3]. Monochromatic translucent zirconia was later introduced. However, they were not sufficient to produce esthetically pleasing zirconia restorations. This was mainly because of the difficulty in achieving color accuracy [4]. Therefore, manufacturers opted to produce a translucent multi-layered color-gradient zirconia material [5]. This unique innovation had the goal of mimicking the color gradient of natural detention [6].

These novel multi-layered color-gradient zirconia materials were uniquely manufactured in a way that allows them to mimic the color gradience of natural teeth. Manufacturers achieved this effect by adding different percentages of metal oxides, such as $Fe_2O_3$, in different layers while maintaining the same microstructure in all the layers [5,7]. As we move from the incisal layer to the cervical layer, the percentage of metal oxide increases

gradually, leading to an increase in chroma at the cervical layer compared to the incisal layer [5]. However, for any dental material to succeed, it must not only be esthetically pleasing but it should also be mechanically stable. The compositional difference in metal oxide incorporation between the layers of multi-layered zirconia could lead to a different mechanical behavior of these materials upon different treatments such as sintering [8,9].

Sintering temperature has been proven to affect the grain growth, phase content, and porosity, which, in turn, can influence the surface microhardness and flexural strength [10,11]. Therefore, the mechanical stability of zirconia materials is significantly affected by the sintering temperature [12]. Sintering zirconia restorations at inadequate temperatures could lead to a deterioration in their mechanical properties, ultimately leading to failure [13].

Sintering temperatures ranging from 1350 °C to 1600 °C have been reported in previous studies [12,14–16]. This wide range could create zirconia restorations with variable flexural strength and surface microhardness values. Therefore, it is crucial to investigate the effect of sintering temperature on the mechanical and surface properties of the newly developed multi-layered zirconia restorations.

The variation in the Vickers microhardness of monochromatic translucent zirconia after using different sintering temperatures was explored in several studies [17,18]. Kaizer et al. investigated the effects of multiple sintering parameters, including temperature, on the hardness of zirconia crowns [18]. They found that the groups sintered at higher temperatures and lower dwell times exhibited lower hardness values compared to the conventionally sintered group [18]. On the other hand, another study found no significant difference in the Vickers surface hardness values of groups sintered at temperatures greater than 1200 °C [17]. The effect of the sintering temperature on individual layers of multi-layered zirconia is scant. One study investigated the effects of the sintering time and temperature on the surface microhardness [19]. They concluded that as the temperature increased and dwell time decreased, the surface microhardness significantly increased [19].

Several studies investigated the effect of changing sintering temperature on the three-point flexural strength of zirconia restorations [14,20]. One study examined the effect of different sintering temperatures (1350, 1450, and 1550 °C) on the flexural strength of zirconia material [21]. The authors found that as sintering temperature increased, flexural strength was improved [21]. In contrast, a recent study showed that changing the sintering temperature on the flexural strength had no significant effect [14]. Öztürk et al. also investigated zirconia and found that a change in sintering temperature did not have a significant effect on the flexural strength of the tested zirconia restorations [20]. However, these studies were conducted on monochromatic zirconia. To our knowledge, no studies investigated the effect of different sintering temperature on multi-layered color-gradient zirconia materials.

Therefore, the aim of this study was to investigate and compare the effects of the sintering temperature on the Vickers microhardness and three-point flexural strength values of two monolithic translucent multi-layered zirconia materials. The null hypotheses included no significant difference in the Vickers microhardness values of the multi-layered zirconia materials, at the same sintering temperature and layer, no significant difference in the flexural strengths of the multi-layered zirconia materials at different sintering temperatures, no significant difference in the Vickers microhardness and flexural strength values at different sintering temperatures, and, finally, no significant correlation between the sintering temperature and either the Vickers microhardness or flexural strength.

## 2. Materials and Methods

### 2.1. Sample Preparation

In total, 96 plate-shaped specimens were fabricated for Vickers microhardness testing from two types of pre-sintered multi-layered zirconia blanks: DD cube ONE ML (4Y-TZP) and DD cubeX$^2$ ML (5Y-TZP). The details of the materials used in the study are listed in Table 1.

**Table 1.** List of materials.

| Material Class | Type | Shade | Name | Manufacturer | Size | Chemical Composition (wt.%) |
|---|---|---|---|---|---|---|
| 5Y-TZP | Multi-layered, Color gradient | A2 | DD cubeX$^2$ ML | Dental Direkt, Spenge, Germany | 98.5 mm × 14 mm | $ZrO_2 + HfO_2 + Y_2O_3 \geq 99.0$ <br> $Y_2O_3 < 10$ <br> $Al_2O_3 \leq 0.01$ <br> Other oxides < 1 |
| 4Y-TZP | Multi-layered, Color gradient | A2 | DD cube ONE ML | Dental Direkt, Spenge, Germany | 98.5 mm × 14 mm | $ZrO_2 + HfO_2 + Y_2O_3 \geq 99.0$ <br> $Y_2O_3 < 8$ <br> $Al_2O_3 \leq 0.15$ <br> Other oxides < 1 |

The multi-layered pre-sintered zirconia blanks were cut into four sections to obtain samples from each color gradient layer using a water-cooled low-speed diamond saw that was 0.5 mm thick (Isomet 2000, Buehler, Lake Buff, IL, USA). The material was sectioned according to the location of each layer provided by the manufacturer. The first layer corresponded to the enamel layer (3.5 mm thick); the second layer was intermediate 1 (2.1 mm thick); the third was intermediate 2 (2.1 mm thick); and the fourth layer was the body/dentin layer (6.3 mm thick). Subsequently, plate-shaped specimens were cut from each layer using the water-cooled low-speed diamond saw. Each fabricated specimen was 25% larger than the required final size to compensate for shrinkage after sintering/crystallization, as well as the finishing and polishing process. Finally, the pre-sintered plates fabricated from each layer of both zirconia materials were divided randomly into three subgroups according to the sintering temperature: 1300, 1450, and 1600 °C. The design of the study is shown in Figure 1.

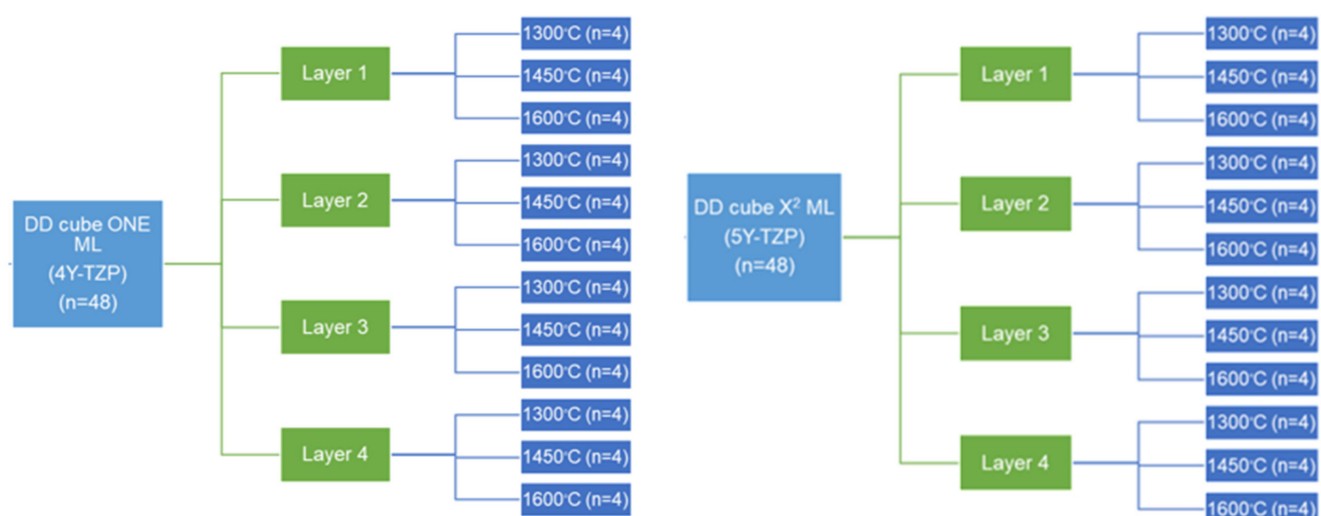

**Figure 1.** Study design and distribution of plate-shaped samples from each layer for Vickers microhardness testing.

Next, 72 bar-shaped specimens were fabricated from the same two types of pre-sintered multi-layered zirconia blocks. Detailed information about these materials is provided in Table 1. These bar-shaped specimens were cut from the full thickness of the zirconia blanks, regardless of the layers, using the same water-cooled low-speed diamond saw. Each fabricated specimen was 25% larger than the required final size to compensate for shrinkage after sintering/crystallization, as well as the finishing and polishing process. The bar-shaped specimens were later divided into three groups according to the sintering temperature: 1300, 1450, and 1600 °C. The design of this study is shown in Figure 2.

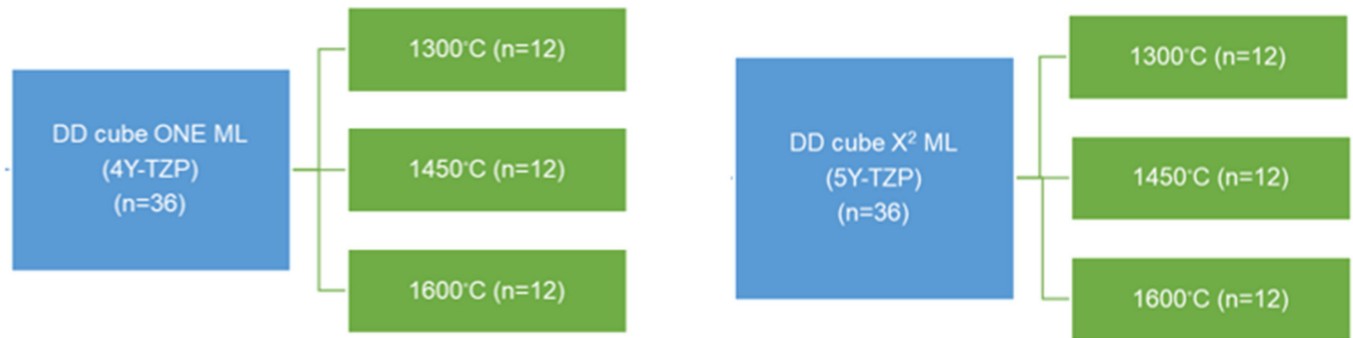

**Figure 2.** Study design and distribution of bar-shaped samples for flexural strength testing regardless of the layers.

All the zirconia samples (plate/bar) were sintered in the same furnace (inFire HTC, Dentsply Sirona, Bensheim, Germany) and with the same holding time (2 h). The plates were ground flat and polished with a series of silicon carbide sheets (600-, 800- and 1200-grit) at 300 rpm under water cooling until a mirror-like surface was achieved, using a standardized finishing and polishing time. The final size of the plate-shaped specimens was $10 \times 10 \times 1$ mm $\pm$ 0.02 mm. On the other hand, the dimensions of the bar-shaped specimens were $16 \times 4 \times 2$ mm $\pm$ 0.02 mm. The definitive thickness of all the samples was determined using a digital caliper (Mitutoyo Digimatic, Mitutoyo Canada Inc., Toronto, ON, Canada), with an accuracy of $\pm$0.05 mm. All the specimens were stored and tested under dry conditions.

### 2.2. Vickers Microhardness

The Vickers microhardness ($kg/mm^2$) of each specimen was measured by the indentation method using a Vickers microhardness tester (NOVA 130 series, Micro Vickers and Knoop hardness testing instrument, Innovatest Europe BV, Maastricht, The Netherlands), which is a square pyramidal indenter. The test was performed under dry conditions at room temperature with a load of 9.8 N applied for 10 s according to the recommendations of ASTM C1327 [22]. Five indentations were made near the center of the specimen with a minimum spacing of 0.5 mm away from each other and the specimen margins. Then, the average was taken as the Vickers microhardness value (HV) for each specimen using the following equation:

$$\text{Vickers microhardness} = 1.8544 \times (P/d^2), \tag{1}$$

where P is the applied load in kilograms and d is the mean diagonal length of the indentation in millimeters.

### 2.3. Three-Point Flexural Strength

The three-point flexural strength tests were conducted on the bar-shaped specimens according to the ISO 6872: 2015 standard [23]. The specimens were tested under dry conditions in a universal testing machine (ElectroPuls E3000, Instron Co., Norwood, MA, USA) at a crosshead speed of 0.5 mm/min until failure. The distance between the supporting beams was 10 mm, and the full scale was 10 kN. The flexural strength (UNI, in MPa) was calculated using the following equation:

$$\sigma = 3Pl/2wb^2, \tag{2}$$

where $\sigma$ is the flexural strength (MPa), P is the fracture load (N), l is the distance between supporting beams (mm), w is width of the specimen (mm), and b is the thickness of the specimen (mm).

### 2.4. Fractogprahic Analysis

First, the fractured fragments from each sample were ultrasonically cleaned for 10 min using distilled water and then air-dried. Next, an overall examination was performed using a low-power digital microscope (DIGITAL MICROSCOPE KH-7700, Hirox, Tokyo, Japan) to determine the fracture origin and fractographic features [24]. After that, the selected representative samples were coated with gold sputter (JFC-1100, JEOL Ltd., Tokyo, Japan) and examined using scanning electron microscopy (SEM) (JSM-6360 LV, JEOL Ltd., Tokyo, Japan) with magnifications ranging from $70\times$ up to $8500\times$ to confirm the fracture origins and locate the fracture features. SEM images of the fracture surfaces were taken, and the fracture origin regions were magnified to characterize the fracture patterns and micromechanism.

### 2.5. Statistical Analysis

Descriptive statistics were first calculated to report the mean, standard deviation, and minimum and maximum values for each subgroup. Before conducting the hypothesis testing, the data were assessed for normality using a Shapiro–Wilk test. Levene's test was conducted to determine the homogeneity of the variances within the groups for each dependent variable. Based on the number of means, either a T-test or one-way analysis of variance (ANOVA) was conducted to evaluate the significant differences between the means.

A T-test was conducted to compare the mean Vickers microhardness and flexural strength values of both zirconia materials at different sintering temperatures. Furthermore, an ANOVA was performed to examine the difference in the means of the outcomes for the Vickers microhardness at different sintering temperatures and layers. An ANOVA was also conducted to evaluate the difference in the mean flexural strengths at different sintering temperatures. When the ANOVA resulted in at least one different group ($p < 0.05$), a post hoc multiple comparison test (Dunnett T3) was used to determine the significant differences between the compared groups.

Finally, the correlation between the sintering temperature and either the Vickers microhardness or flexural strength was assessed using the Pearson correlation coefficient ($r$). Data were analyzed using SPSS 24.0 software, with a $p$-value < 0.05 (IBM Inc., Chicago, IL, USA).

## 3. Results

### 3.1. Vickers Microhardness

The results of the T-test comparing the two materials at the same sintering temperature, regardless of the layers, are seen in Table 2. It was found that 4Y-TZP had a significantly higher microhardness than 5Y-TZP at all sintering temperatures ($p < 0.05$) (Table 2).

**Table 2.** Results of the T-test comparing the mean flexural strength and Vickers microhardness of zirconia materials at the same sintering temperature.

| Measurement | Sintering Temperature (°C) | Zirconia Material | Mean $\pm$ SD |
|---|---|---|---|
| Microhardness (kg/mm$^2$) | 1300 °C | 4Y | 1306.57 $\pm$ 17.93 [a] |
| | | 5Y | 1269.48 $\pm$ 15.95 [b] |
| | 1450 °C | 4Y | 1468.87 $\pm$ 35.11 [a] |
| | | 5Y | 1399.59 $\pm$ 30.55 [b] |
| | 1600 °C | 4Y | 1398.92 $\pm$ 28.64 [a] |
| | | 5Y | 1354.86 $\pm$ 44.0 [b] |
| Flexural strength (MPa) | 1300 °C | 4Y | 758.66 $\pm$ 13.21 [a] |
| | | 5Y | 372.28 $\pm$ 12.43 [b] |
| | 1450 °C | 4Y | 1390.55 $\pm$ 28.61 [a] |
| | | 5Y | 669.58 $\pm$ 22 [b] |
| | 1600 °C | 4Y | 1157.48 $\pm$ 17.82 [a] |
| | | 5Y | 500.97 $\pm$ 26.26 [b] |

Different superscript letters indicate a significant difference ($p < 0.05$) between the means within the same column.

Furthermore, at 1300 °C, a significant difference was found between both zirconia materials in all the layers ($p < 0.05$) (Figure 3). However, at 1450 °C, the only significant difference detected between the two zirconia materials was found at layers 3 and 4 and at layer 1 when sintered at 1600 °C ($p < 0.05$) (Figure 3). In all the cases evaluated, 4Y-TZP was found to have the highest microhardness value.

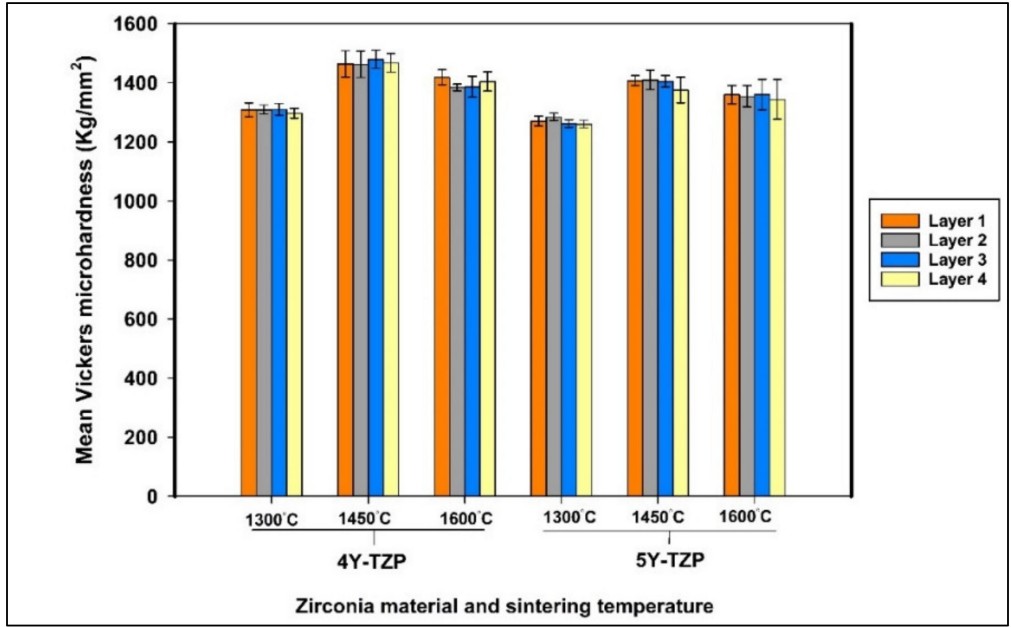

**Figure 3.** Mean Vickers microhardness of both zirconia materials at different layers and sintering temperatures.

When evaluating the trend of the microhardness values for the different sintering temperatures, a significant improvement in microhardness was found when the sintering temperature increased to 1450 °C ($p < 0.001$) (see Table 3). However, the values significantly decreased when the temperature was raised up to 1600 °C ($p < 0.001$) (Table 3).

**Table 3.** Results of one-way ANOVA comparing the average Vickers microhardness of all the layers and flexural strength at different sintering temperatures followed by multiple comparison test (Dunnett T3).

| Measurement | Sintering Temperature (°C) | Mean $\pm$ SD | *p*-Value |
| --- | --- | --- | --- |
| Microhardness (kg/mm$^2$) | 1300 °C | 1288.02 $\pm$ 25.17 [a] | 0.000 |
| | 1450 °C | 1434.23 $\pm$ 47.81 [b] | |
| | 1600 °C | 1376.89 $\pm$ 42.83 [c] | |
| Flexural strength (MPa) | 1300 °C | 565.47 $\pm$ 12.82 [a] | 0.000 |
| | 1450 °C | 1030.07 $\pm$ 25.3 [b] | |
| | 1600 °C | 829.22 $\pm$ 22.04 [c] | |

Different superscript letters indicate a significant difference ($p < 0.05$) between the means within the same column.

The results of the one-way ANOVA comparing the mean microhardness values of zirconia layers within the same material and at the same sintering temperature, followed by a multiple comparison test (Dunnett T3), are listed in Table 4. No significant difference was detected among the layers of the same material at the same sintering temperature ($p > 0.05$) (see Table 4).

**Table 4.** Results of one-way ANOVA comparing the mean Vickers microhardness of zirconia layers within the same material and sintering temperature followed by multiple comparison test (Dunnett T3).

| Measure-ment | Zirconia Material | Layer | Mean ±SD at Different Layers and Sintering Temperatures (°C) | | | | | |
|---|---|---|---|---|---|---|---|---|
| | | | 1300 °C | *p*-Value | 1450 °C | *p*-Value | 1600 °C | *p*-Value |
| Microhar-dness (kg/mm$^2$) | 4Y | 1st Layer | 1308.61 ± 23.49 [a] | | 1464.54 ± 44.41 [a] | | 1419.35 ± 25.12 [a] | |
| | | 2nd Layer | 1309.59 ± 15.5 [a] | 0.727 | 1463.19 ± 44.93 [a] | 0.917 | 1384.73 ± 11.37 [a] | 0.227 |
| | | 3rd Layer | 1310.75 ± 19.74 [a] | | 1480.04 ± 30.77 [a] | | 1386.97 ± 35.26 [a] | |
| | | 4th Layer | 1297.31 ± 16.45 [a] | | 1467.70 ± 31.44 [a] | | 1404.62 ± 32.21 [a] | |
| | 5Y | 1st Layer | 1270.30 ± 16.85 [a] | | 1407.40 ± 16.51 [a] | | 1360.14 ± 30.44 [a] | |
| | | 2nd Layer | 1285.0 ± 12.28 [a] | 0.133 | 1410.35 ± 33.3 [a] | 0.661 | 1354.57 ± 36.56 [a] | 0.978 |
| | | 3rd Layer | 1262.14 ± 13.13 [a] | | 1404.92 ± 19.48 [a] | | 1360.50 ± 51.73 [a] | |
| | | 4th Layer | 1260.5 ± 12.81 [a] | | 1375.70 ± 43.26 [a] | | 1344.23 ± 34.15 [a] | |

Different superscript letters indicate a significant difference ($p < 0.05$) between the means within the same column.

### 3.2. Three-Point Flexural Srength

The results of a T-test comparing the mean flexural strength values of both multi-layered materials at different sintering temperatures are listed in Table 2. The results show that 4Y-TZP had a significantly higher flexural strength than 5Y-TZP at all the sintering temperatures ($p < 0.001$).

The results of the one-way ANOVA comparing the mean flexural strength values at different sintering temperatures, followed by a multiple comparison test (Dunnett T3), are listed in Table 3. A significant difference was found between the values at all the sintering temperatures ($p < 0.001$). The highest values were found when the sintering temperature was 1450 °C, followed by those at 1600 °C and, finally, 1300 °C.

The Pearson correlation coefficient (*r*) was calculated to evaluate the correlation between the sintering temperature and either the microhardness or flexural strength. The results showed a significant positive Pearson correlation coefficient ($r = 0.505$) between the sintering temperature and microhardness ($p < 0.001$). Similarly, a significantly positive Pearson correlation coefficient ($r = 0.465$) was found between the sintering temperature and flexural strength ($p < 0.001$).

### 3.3. Fractogprahic Analysis

The two types of zirconia materials had similar fracture patterns at different sintering temperatures. In addition, similar fracture features were observed when comparing the SEM images of the two types of zirconia materials at the same sintering temperature (Figures 4 and 5). Such features included compression curls, hackle lines, and crack arrest lines. However, distinct features were visible at different sintering temperatures for both materials.

SEM images of 4Y-TZP and 5Y-TZP samples sintered at 1450 °C showed typical fracture features of zirconia samples (Figures 4a,b and 5a,b). The features detected included the fracture origins, hackles, compression curls, and arrest lines. At a sintering temperature of 1300 °C (Figures 4c,d and 5c,d), both materials had a relatively smooth fractured surface. This group had the fewest fracture features out of all the sintering temperatures. The features detected included pitting, twist hackles, and compression curls. Finally, zirconia samples from both types of materials sintered at 1600 °C were examined (Figures 4e,f and 5e,f). This group showed increased roughness, irregularities, and pitting.

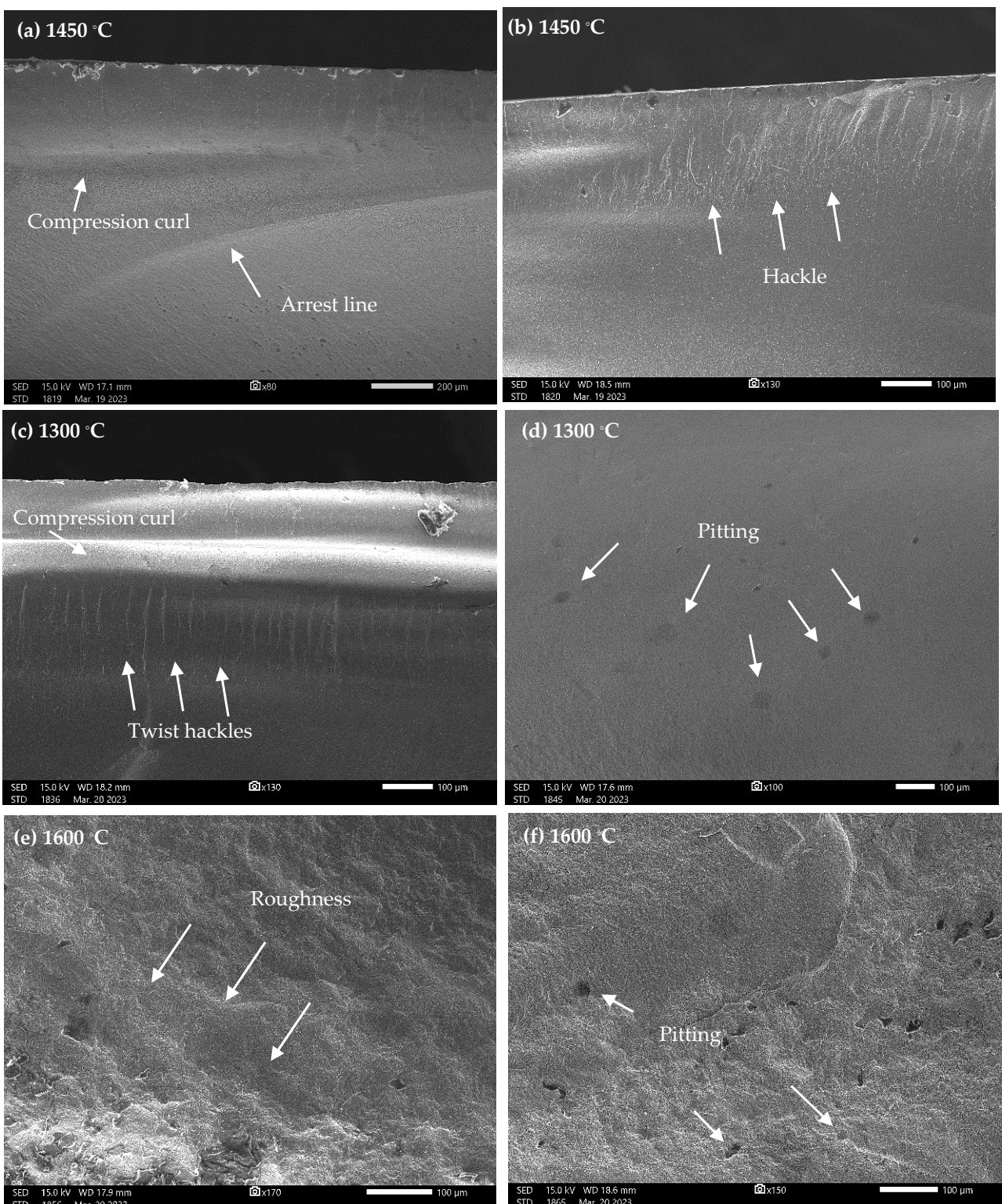

**Figure 4.** Representative SEM images of DD cube ONE ML (4Y-TZP) at different sintering temperatures. (**a**) A sample sintered at 1450 °C showing a compression curl along with an arrest line. (**b**) Higher magnification shows the hackle lines. (**c**) A sample sintered at 1300 °C showing a compression curl along with twist hackles. (**d**) Increased number of pitting is also seen in samples sintered at 1300 °C. (**e**) A sample sintered at 1600 °C showing increases in surface roughness. (**f**) Pitting can also be seen in samples sintered at 1600 °C.

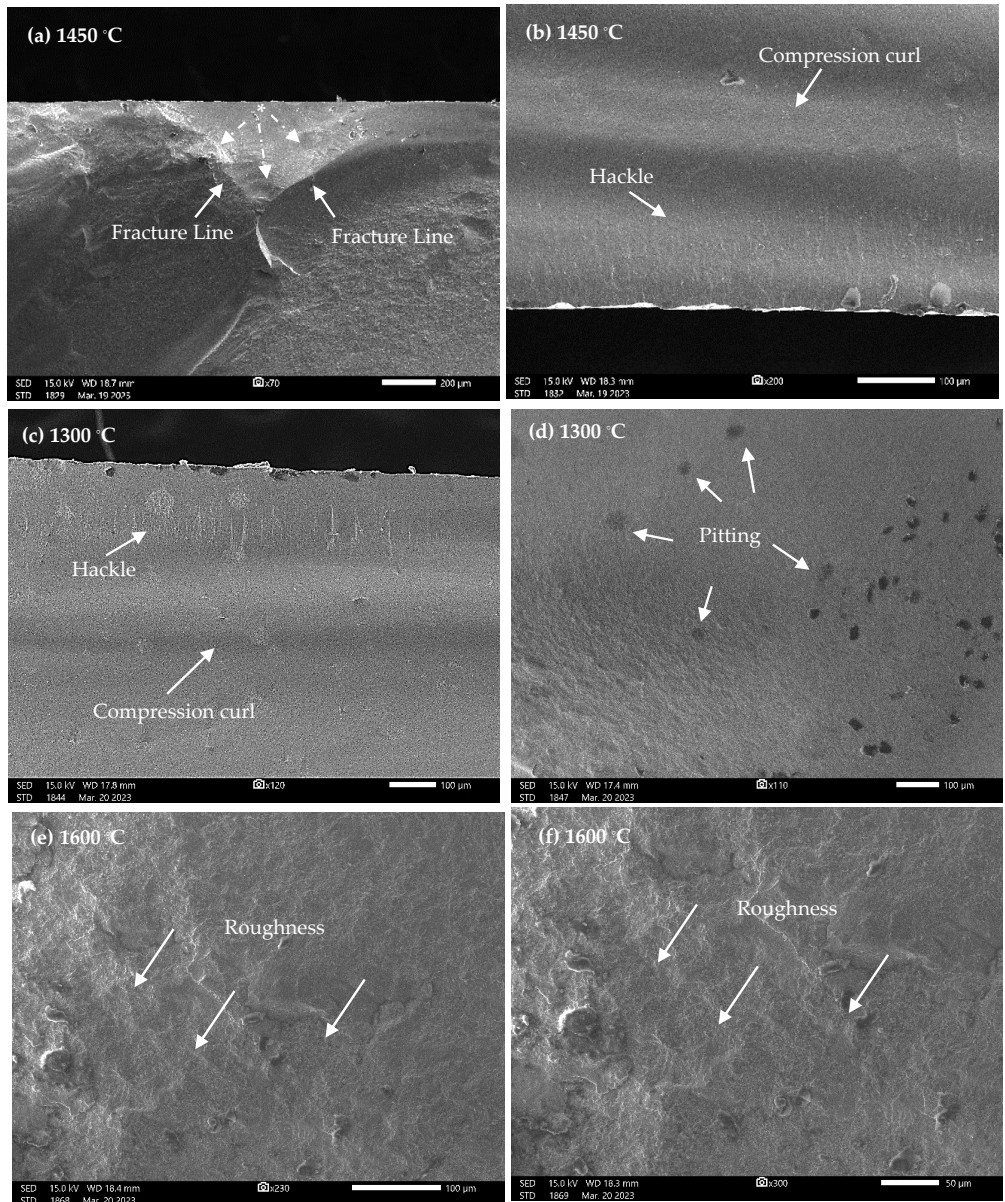

**Figure 5.** Representative SEM images of DD cubeX$^2$ ML (5Y-TZP) at different sintering temperatures. (**a**) A sample sintered at 1450 °C showing the origin of fracture (asterisk) and direction of crack propagation (dotted arrows) along with fracture lines. (**b**) A higher magnification of the same sample shows a compression curl along with hackle lines on the opposite side of the fracture origin. (**c**) A sample sintered at 1300 °C showing a smooth facture surface with a compression curl and hackle lines. (**d**) Increased amount of pitting is also seen in samples sintered at 1300 °C. (**e**) A sample sintered at 1600 °C showing increases in surface roughness. (**f**) Surface roughness and increased irregularities are seen in samples sintered at 1600 °C.

The fracture micromechanism can be identified by the characteristic features apparent in their microstructure visible in the SEM images (Figure 6). There are three mechanisms of fracture that can be identified in the SEM image (Figure 6). The first micromechanism identified is related to samples that have been sintered at 1300 °C (Figure 6a). In this group, clusters of partially sintered grains are seen along with increased voids apparent between them. This weak microstructure allows for the crack to propagate between the clusters without cleavage, leading to an intergranular mode of fracture. The second micromechanism identified is related to specimens sintered at 1450 °C (Figure 6b). In this group, agglomerates of fine grains that are highly cohesive are seen along with the presence

of a few cleaved grains. These features indicate the occurrence of both transgranular and intergranular fracture, meaning that the path of fracture passed along the grain boundaries of the highly cohesive agglomerates along with cleaving a number of grains. The third micromechanism is related to the group sintered at 1600 °C (Figure 6c). The microstructure visible in the SEM images for this group reveals a coarse highly cohesive microstructure with intense growth in grains. Further, multiple cleaved grains are seen in this group, meaning that the fracture propagated through these large grains, leading to the occurrence of predominantly transgranular fracture mode.

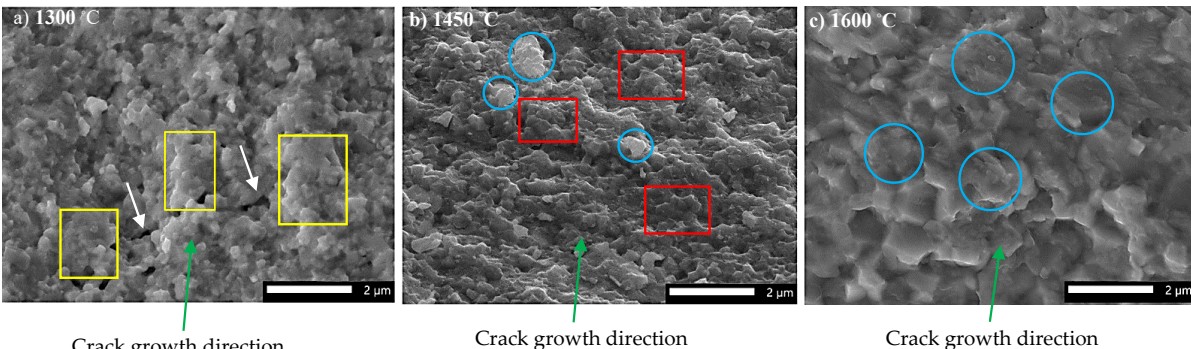

**Figure 6.** Representative SEM images of the fractured surfaces of zirconia samples, sintered at different sintering temperatures, showing their microstructure with characteristic features related to distinct facture micromechanisms. The crack growth direction is also indicated (green arrow). (**a**) At 1300 °C, the microstructure reveals multiple clusters of partially sintered particles (yellow box) along with the presence of large pores (white arrow) that can facilitate crack propagation. (**b**) At 1450 °C, the microstructure appears highly cohesive with the presence of multiple agglomerates of fine grains (red box) with the occasional appearance of cleaved grains (blue circle). (**c**) At 1600 °C, the SEM images reveal an intense grain growth with a coarse and highly cohesive microstructure along with the appearance of intense cleavage of the large grains (blue circle).

## 4. Discussion

The Vickers microhardness of 4Y-TZP was higher than that of 5Y-TZP at all the sintering temperatures. When comparing the layers of both materials, again, 4Y-TZP recorded a higher microhardness value. Therefore, the null hypothesis was rejected. These results were in agreement with previous investigations [11]. This can be explained by the compositional differences between the two zirconia types. The decreased cubic phase content in 4Y-TZP compared to 5Y-TZP leads to higher mechanical and surface properties [25].

A significant difference in the Vickers microhardness values was found at different sintering temperatures. Therefore, the null hypothesis was rejected. The lowest Vickers microhardness values were found at a sintering temperature of 1300 °C. At low sintering temperatures, similar results were recorded by Amat et al. [26]. They found that at low sintering temperatures, grains were not fully densified. There was greater microporosity and, therefore, a weak microstructure, leading to low microhardness values [26].

In our study, increasing the sintering temperature to 1450 °C led to a significant enhancement in microhardness. This could have been due to the increased density of the zirconia specimens, leading to a pore reduction and enhanced crystalline arrangement and, therefore, enhanced microhardness [17,27]. However, increasing the sintering temperature to 1600 °C led to decreases in the microhardness values of both zirconia materials. It was found that increasing the sintering temperature beyond certain limits could increase the grain size of the zirconia material, leading to a decrease in microhardness [18].

When measuring the Vickers microhardness values of the layers within the same zirconia material and sintering temperature, no significant difference was found between the layers. Therefore, the null hypothesis was rejected. These findings were in agreement with previous results [28]. They evaluated the surface characteristics of different layers of

multi-layered translucent zirconia materials and found no significant difference between the tested layers. This was explained by the fact that different layers within the same material had similar yttria and crystal-phase contents. Hence, they had similar microstructures, leading to similar Vickers hardness values. The only difference found between the layers was in the pigment composition [5,28].

The three-point flexural strength test resulted in significantly higher values for 4Y-TZP compared to 5Y-TZP at all the sintering temperatures. Therefore, the null hypothesis was rejected. The higher values observed for 4Y-TZP compared to 5Y-TZP were expected because of the microstructural differences between them. Sulaiman et al. conducted a flexural strength test followed by a microstructural analysis of partially and fully stabilized monolithic zirconia materials [29]. They found that the fully stabilized zirconia recorded lower flexural strength values, even after different treatments compared to the partially stabilized zirconia. A microstructural analysis also revealed higher yttria and cubic-phase contents with a larger grain size compared to the partially stabilized zirconia [29].

It has been proven that the mechanical properties of crystalline ceramics are highly dependent on their microstructure [29,30]. The increased yttria content present in 5Y-TZP leads to an increase in the cubic phase, accompanied by a decrease in the tetragonal-phase content [31–33]. A decrease in the *t*-phase, which is well known for its high mechanical strength compared to the other three crystallographic forms, will lead to poor mechanical properties, including flexural strength [13].

Furthermore, in our study, it is noticeable that 4Y-TZP had a remarkable response to different sintering temperatures compared to 5Y-TZP. This result was in accordance with a previous study [13]. In their study, they found that the variation in flexural strength values of 5Y-TZP at different sintering temperatures were negligible compared to 4Y-TZP [13]. This can be explained by the microstructural differences between the two zirconia types. The percentage of *t*-phase, responsible for the increased flexural strength, is initially significantly less than that present in the 4Y-TZP microstructure. It was reported that the highest percentage of *t*-phase present in the microstructure of 5Y-TZP was only 29%, while 4Y-TZP had almost 40% [13]. Therefore, the changes in *t*-phase percentage after different sintering temperatures were negligible compared to other highly abundant *t*-phase zirconia materials [13].

The flexural strength values recorded at all the sintering temperatures were significantly different. Therefore, the null hypothesis was rejected. The lowest values were recorded at 1300 °C, followed by those at 1600 °C and, finally, 1450 °C. These results were in accordance with previous findings [12,26,34]. Sintering parameters can affect the grain growth, porosity, and phase composition, and, therefore, affect the mechanical properties of the material [33,35]. Most zirconia restorations are sintered at temperatures above 1350 °C [14,16]. At lower temperatures such as 1300 °C, lower values of flexural strength are expected as a result of the formation of immature grains, along with an increase in voids, which create a material with lower density that is more susceptible to breakage [26].

Significantly higher flexural strength values were recorded at the manufacturer-recommended temperature of 1450 °C. Vult von Steyern et al. also found that when sintering zirconia materials at the recommended temperature, the flexural strength values were the highest compared to other tested temperatures [13]. This result can be justified by an observed adequate grain growth, an increase in the tetragonal-phase content, along with the formation of a material with higher density and lower voids [13].

Lastly, it was found that at 1600 °C, the flexural strength values significantly dropped from their values at 1450 °C. Our findings agree with those of previous studies [12,26]. Sintering zirconia at higher temperatures creates thermal stresses that can over-induce the *t-m* transformation, resulting in a significant increase in the grain volume and leading to grain pullout, increased roughness, voids, and the development of microcracks [12,26]. These changes were proven to have a negative effect on the flexural strength [12,26]. On the other hand, a study found no significant change in the flexural strength of the tested zirconia material when sintered at higher sintering temperatures [13]. This was justified by the fact that one of the materials utilized in their study was 5Y-TZP, which showed a

slight decrease in the *t*-phase content compared to other types of zirconia materials such as 4Y-TZP. This decrease would not have a significant impact on the initially deficient tetragonal-phase content observed in 5Y-TZP materials [13].

The Pearson correlation coefficient between the sintering temperature and either the Vickers microhardness or flexural strength was significant. Therefore, the null hypothesis was rejected. Both the microhardness and flexural strength had significant positive correlations with the sintering temperature. In other words, these values increased with the sintering temperature. This was mainly due to the direct effect of the sintering temperature on the grain growth, density, and phase composition [36]. However, excessively increasing the sintering temperature can decrease the mechanical stability of zirconia materials.

The SEM investigation found that when comparing the 4Y-TZP and 5Y-TZP at the same temperature, similar fractographic features were found. However, distinct features were found when samples were sintered at different temperatures. At 1450 °C, typical zirconia fracture features were detected on the fracture surface. At 1300 °C, on the other hand, minimum fractographic features were seen with increased pitting. Finally, at 1600 °C, increased roughness was noticed. Similar features were noted in previous studies that evaluated the effects of different sintering modes and temperatures on zirconia materials [37,38].

Kulyk et al. investigated the effects of different sintering modes on the fracture mechanism of zirconia materials using SEM [37]. They reported three different mechanisms of facture related to three different sintering temperatures. They found that when the sintering mode promoted complete crystallization, such as when sintering at the recommended temperature of 1450 °C, the SEM images revealed a fine-grained, highly cohesive microstructure with a minimum number of voids. This microstructure promoted high strength because of the ability of the mature grains to withstand crack propagation.

This was confirmed by the mechanism of fracture seen in the SEM images showing the propagation of the crack along the fine grains, with cleavage taking place occasionally through the grains that were located at the path of the fracture [37]. These features are also apparent in our SEM images (Figure 6b), leading to the occurrence of both transgranular and intergranular fracture modes. This fracture micromechanism is considered to be the most favorable fracture mechanism due to the multiple branching that occurred in the crack path, leading to a deflection in the direction of the crack growth [37]. This deflection allows for the speed of fracture propagation to decrease, therefore, improving the strength and toughness of the material [37].

The second fracture mechanism was related to sintering modes that did not promote adequate crystallization, such as low sintering temperatures. They found that such zirconia materials had the lowest fracture resistance because of the weak coherence of the grains, increased size of the gaps between the grains, and increased number of pores. These microstructural features, also seen in Figure 6a, allow for the occurrence of mostly intergranular fracture mode. This comparatively weak microstructure led to a significantly low resistance to crack propagation, with the pores and inter-grain spacing creating a path for rapid crack growth [37,38]. This could explain the low fractographic features seen in our study when the sintering temperature was 1300 °C (Figures 4c,d and 5c,d).

Finally, the third fracture mechanism observed in the study conducted by Kulyk et al. was related to much higher sintering temperatures. It was found that the crack growth was more severe than that in the previous two fracture mechanisms. This was linked to the microstructure seen in this group. An increased sintering temperature led to the formation of intense grain growth and coarse grain boundaries and, therefore, increased roughness (seen in Figures 4e,f and 5e,f). This microstructure caused the crack growth to progress within the intensely large grains, leading to transgranular fracture mode characterized by the splitting of these grains, rather than going around the boundaries [37,38]. Therefore, mostly cleaved grains are seen in the SEM images (Figure 6c).

The limitations of our study include only comparing two types of zirconia materials and focusing on the mechanical properties. Future studies should include more materials, other physical and optical properties, and different sintering parameters.

## 5. Conclusions

Within the limitation of this study, it can be concluded that mechanical properties, including the flexural strength and Vickers microhardness, can be determined by the type of zirconia material, along with the sintering temperature. However, the different layers of the two materials did not have a significant effect on the microhardness values. This could have been because the same microstructure was seen in all the layers of the same type of material. The 4Y-TZP had higher flexural strength and Vickers microhardness values than 5Y-TZP at all the layers and sintering temperatures. Furthermore, the flexural strength and Vickers microhardness values were found to be the highest when the sintering temperature was 1450 °C, followed by those at 1600 °C and, finally, 1300 °C. The SEM images revealed features that indicated the occurrence of both transgranular and intergranular fracture modes at 1450 °C, intergranular fracture mode at 1300 °C, and, finally, transgranular fracture mode at 1600 °C.

**Author Contributions:** Conceptualization, methodology, software, validation, formal analysis, investigation, data curation, writing—original draft preparation, B.A. and A.A.-A.; resources, B.A.; writing—review and editing, B.A.; supervision, A.A.-A.; project administration, B.A.; funding acquisition, B.A. All authors have read and agreed to the published version of the manuscript.

**Funding:** No funding was received for this study.

**Institutional Review Board Statement:** Not applicable.

**Informed Consent Statement:** Not applicable.

**Data Availability Statement:** Data sharing is not applicable to this article.

**Acknowledgments:** This study was registered and approved by the College of Dentistry Research Center (registration number: PR 0118). This manuscript is part of a DScD dissertation.

**Conflicts of Interest:** There is no known conflict of interest associated with this study, and there has been no significant financial interest in any company, or any products mentioned in this manuscript that could have influenced the outcome.

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
