# Peer review of "The Effect of Sintering Temperature on Vickers Microhardness and Flexural Strength of Translucent Multi-Layered Zirconia Dental Materials"

_coatings, doi:10.3390/coatings13040688_

Round 1

Reviewer 1 Report

Reviewer Comment

Manuscript Number: Coatings-2298231

Dear Editor,

The effect of sintering temperature was investigated on both hardness and flexural strength of ZrO2 used for dental restorations. The introduction section is written with a good flow which ensured a well understanding while this trend was missing in the Results section.

1-The last paragraph in the introduction section, line 66 onward, is not clear. The authors should clarify it and support it by references.

2-Table 1, chemical formulas of compounds should have a correct subscription  such as Y2O3, HfO2, ZrO2, Al2O3, ..

3-Technical error was observed in Figure 1. Overlapping between sentence orders and the figure itself.

4-Table 2, Kg/mm2  should be corrected kg/mm2.

5-Table 2 and Figure 3 are refer to the same evaluation. What is the purpose to provide figure and table for the same representation?

6-the statement from line 211 onwards, contradicts what it is shown in figure 3. at 1300 C, all layers have almost the same hardness value!

7-line 264: Pearson correlation ...

8-Is the Table 3 Mean SD values for hardness and flexural strength for ZrO2 materials regardless of the layers? or the values are averages among all layers to obtain a certain value for each sintering temperature for their corresponding mechanical property?

That is not clear from the caption of the Table 3.

9- ◦C is the correct format for temperature

10- the Table 4 ! 4th layer has a significant difference in value for flexural strength. The authors have not mentioned it.

11-what is r2 and p indicate? the authors did not mention to these terms and how these terms used to evaluate the significance of models. They also did not provide any references to support their ideas.

12-Line 459-461: is totally opposite. Transgranular fracture mode (which crack path is through grains) has higher fracture toughness than intergranular fracture mode (which crack path is along grain boundaries) due to the higher energy consumption by the former.

https://doi.org/10.1016/j.ceramint.2013.11.076 

The reference that authors cite it, it explain that agglomeration of fine grains occur which lead to taking place of cleavages, and the crack path though several of these cleavages beside intergranular fracture mode. Means that transgranular fracture mode was already happened! as shown in the figure 13 in the reference
https://doi.org/10.3390/ma15155212

13-Figure 4 and figure 5 of SEM micrographs need higher magnification scales to identify the features of fracture. The fracture modes are not clear.

Reviewer 2 Report

In this manuscript, authors choose two materials DD cube ONE ML (4Y-TZP), DD cube X2 ML (5Y-TZP) as investigation objects to explore the effect of sintering temperature on their Vickers microhardness and flexural strength  and attempt to get a relationship between temperature and microhardness and flexural strength. The study has a certain interest and significance. I would like to recommend this manuscript for publication in Coatings after minor revision.

1. In introduction section, more references relative to present work should be cited and the significance and novelty of this study should be added.

2. What is the intrinsic difference of 4Y-TZP and 5Y-TZP? What is main composition of every layer?

3. How to explain the effect of sintering temperature is not available for every layer in two different materials?

4. What is scientific significance of this study for its application in the future based on these results? 

5. The effect of temperature is quite remarkable for 4Y-TZP compared to 5Y-TZP, authors need to base on their composition to explain this effect in detail. Obviously, different metal oxides should be sensitive for sintering temperature.

Reviewer 3 Report

The authors have studied the effect of sintering temperature on Vickers microhardness and flexural strength of translucent multi-layered zirconia dental materials. The research is well designed and presented clearly. A good comparative analysis of existing publications concerning the tasks set in the work is performed. The methodological section of the manuscript is presented in sufficient detail. The authors used the modern equipment for test of specimens as well as visualization and assistance in the interpretation of the obtained results. They found that 4Y-TZP recorded higher flexural strength and Vickers microhardness values compared to 5Y-TZP at all layers and sintering temperatures. Furthermore, flexural strength and Vickers microhardness values were found to be the highest when sintering temperature was 1450°C followed by 1600°C and finally 1300°C.

However, some shortcomings should be corrected to make the manuscript acceptable for publication in Coatings.

(1) In Sub-section 2.1, the authors should indicate pre-sintering temperature for the ceramics and the approximate thickness of each layer.

(2) In Table 1, the column “Chemical Composition”, the units should be indicated (wt% or others).

(3) In my opinion, English language of this manuscript should be significantly improved.

Round 2

Reviewer 1 Report

Reviewer Comment

Manuscript Number: Coatings-2298231

Dear Editor,

The corrections made by authors made it satisfactory to be published in your esteem journal.